# Self-Testing as a Hope to Reduce HIV in Transgender Women—Literature Review

**DOI:** 10.3390/ijerph19159331

**Published:** 2022-07-30

**Authors:** Julia Budzyńska, Rafał Patryn, Ilona Kozioł, Magdalena Leśniewska, Agnieszka Kopystecka, Tomasz Skubel

**Affiliations:** 1Students’ Scientific Group on Medical Law, Department of Humanities and Social Medicine, Medical University of Lublin, 20-059 Lublin, Poland; ilona.koziol9@gmail.com (I.K.); magdalenagim16@gmail.com (M.L.); aga.kopystecka@gmail.com (A.K.); tomasz.wojciech.skubel@gmail.com (T.S.); 2Department of Humanities and Social Medicine, Medical University of Lublin, 20-059 Lublin, Poland; rafalpatryn@umlub.pl

**Keywords:** HIV, transgender, sexual and gender minority, selftest, marginalization, stigma

## Abstract

So far, the rate of HIV-positive people who do not know their sero-status is about 14% and the percentage is higher among transgender women (TGW). They represent one of the most vulnerable groups to infection. HIV self-testing (HIVST) may be a way to reduce transmission of the virus. The aim of this analysis and in-depth review was to collect available data on factors that may influence the use and dissemination of HIVST among TGW. This review was conducted in accordance with PRISMA guidelines for systematic reviews and meta-analyses. All data from 48 papers were used. From the available literature, HIVST is a convenient and preferred method of testing due to its high confidentiality and possibility of being performed at home. However, there are barriers that limit its use, including marginalization of transgender people, stigma by medical personnel, lack of acceptance of sexual partners, and even cultural standards. Therefore, there is a need for activities that promote and inform on the possibility of using HIVST as well as enable easier access to it.

## 1. Introduction

HIV is a widespread pathogen with more than 5000 daily infections worldwide [1]. Transgendered women (TGW) are particularly vulnerable. They are a diverse group of people who were assigned male sex at birth but mostly identify as women, trans women, and/or transfeminine. They are estimated to be more than 48 times more likely to be infected with HIV than cis women (those whose gender identity matches that assigned at birth) of reproductive age [2,3,4,5,6,7,8,9,10]. Transgender women of color in particular face high rates of infection and low use of important HIV prevention tools [10,11]. A randomized study conducted by Mujugira et al. that included a group of high-risk HIV-infected sex workers including TGW found that more than 80% of respondents feared HIV infection [12]. The number of youths living with HIV in the United States (US) is increasing, and the epidemic is growing among racial, ethnic, and sexual minorities. Reports show that 69% of new infections occur in young men who have sex with men (MSM) and young TGW, and the epidemic is more severe among black and Hispanic youth [13,14,15,16,17,18,19]. HIV prevalence among TGW is estimated to be approximately 14.2%, which is higher compared to other vulnerable populations such as gay, bisexual, and other men who have sex with men (10.6%) [19,20,21]. Additionally, a similar trend is observed for TGW in the Middle East and North Africa, where infection rates continue to rise [5,9]. However, results from a study in São Paulo, Brazil, indicate an even higher prevalence of HIV among TGW compared to data from around the world [2,22]. The estimated prevalence of HIV infection in TGW worldwide is approximately 19.1% [2].

Attempts are made around the world to combat the HIV epidemic through increased emphasis on the prevalence of HIV testing, especially among high-risk populations [23]. U.S. Centers for Disease Control and Prevention and WHO guidelines recommend that people at increased risk for AIDS be tested for HIV infection at least once a year, and in some situations every three to six months [12,24,25,26]. There are several ways to test for HIV infection, such as traditional testing performed at a clinic, self-testing without visiting a physician or clinic, and HIV testing and counseling for couples [27]. HIV self-testing (HIVST) involves a person being able to take a rapid diagnostic test in private, without a third party, which makes it convenient and can provide discretion. Rapid tests are made from a patient’s saliva and, because of their high sensitivity and specificity, have been approved for individual use by the U.S. Food and Drug Administration (FDA) due to their potential to diagnose new cases of HIV [23,28]. A screening test should be sensitive and specific and, above all, easy to perform. It is important to promote such testing among vulnerable groups to raise awareness and ensure their safety. This is the aim of UNAIDS 95-95-95, where 95% of people living with HIV should be aware of it, 95% of these people should take antiretroviral therapy, and even 95% with this therapy should achieve virological suppression by 2030 [29]. For example, in one of the studies it was observed that a small number of participants in a study conducted in Bangkok who tested HIV-positive were aware of their HIV infection. This finding highlights the need for greater access and more frequent HIV testing, especially in key populations such as TGW [24]. However, a global review of the latest literature shows that there is still a lack of European evidence on transgender people and their HIVST [29].

The objective of the review was to identify the factors influencing the popularity of self-testing, which set the limits and directions for the future spread of HIVST diagnostics in TGW. The question also arises as to whether HIVST, by virtue of its characteristics, can contribute to the reduction of HIV prevalence.

## 2. Material and Methods

This review was conducted in accordance with PRISMA guidelines for systematic reviews and meta-analyses (Figure 1). The literature review was conducted in May 2022 using PubMed, Scopus, Web of Science databases with English language restriction, and publications between 2020 and 2022. The keyword combinations used in all databases included: “HIV”, “transgender”, “transgender women”, “transwomen”, “transsexual”, “test”, “selftest”. A study was considered if it presented primary data (qualitative or quantitative) for transgender women and also included information on the following criteria: willingness to use HIVST, and/or advantages and/or limitations of HVIST, future directions related to HVIST. One of the barriers to acceptability was that only pre-exposure prophylaxis was described, and no separate group was identified of which only TGWs were included. Texts that only concerned transgender men were an exception criterion. Commentaries, letters, and editorials were also excluded.

Electronically searched papers were analyzed for eligibility in terms of title and abstract. Duplicates were then eliminated; potentially compatible papers were further read in their entirety. The reasons for exclusion were documented. The analysis was conducted by two independent researchers. The search strategy resulted in a total of 361 papers. After removing duplicates and meeting the criteria, 56 papers were obtained, of which 48 papers were included in the review and bibliographic data were analyzed.

## 3. Results

The search strategy resulted in a total of 361 papers (Figure 1). After removing duplicates and meeting the criteria, 56 papers were obtained, of which 48 papers were included in the review and bibliographic data were analyzed.

### 3.1. Details of the Studied Population

The reports studied were different in terms of the patient population. The review focuses on transgender women, a small group of people with high HIV prevalence and increased stigma in society [2]. However, many of the following articles consider transgender women and men, which is one of the limitations of this study. Furthermore, there are no reports of the use of HIVST on European, including Polish transgender women.

### 3.2. Definition and Types of Tests

HIV self-testing (HIVST) has emerged in recent years as a promising risk reduction strategy. It aims to reduce the HIV epidemic [30]. In particular, it aims to show that HIVST is an easy, convenient, and highly acceptable testing alternative for a variety of key populations around the world [30,31]. This includes populations that may not have access to safe and affordable testing services, including TGW who do not consistently use protection during intercourse [30,32,33,34,35,36,37,38,39]. For such TGW, HIVST kits can be used as an active form of risk reduction.

A quick, home-based HIV test involves collecting oral fluid to detect the presence of HIV antibodies. Individuals swab their gums with the testing device and place the secretion-soaked tip into the prepared solution. After waiting about 20 min, the result can be read [40]. The research findings suggest that the oral fluid-based HIVST test is preferred over blood-based tests [34,41,42].

One of the testing systems developed is HemaSpot-HF. It contains a small plastic device, and you may use it yourself at home to perform the test. After collecting blood (3–5 drops) with a disposable safety puncture device, the HemaSpot-HF can be closed and sent immediately as the desiccant dries the sample inside the cartridge. Study participants sent completed kits directly to the research laboratory using a prepaid return envelope. HemaSpot-HF was developed to solve the technical problems associated with using traditional filter cards to collect a dry blood spot [43]. Studies have shown that men living with HIV are willing and able to collect their own blood samples using HemaSpot-HF and may prefer this option over collecting blood during clinic visits because it provides convenience and privacy [43,44,45].

The availability of HIVST kits has been shown to contribute to more frequent partner testing [30,33,35,46,47]. Currently, little is known about the ways in which TGWs communicate with potential sexual partners about HIVST. Research suggests that offering HIVST to sexual partners may be more effective when kits are introduced in the context of dual or couples testing [30,33,38,47,48,49].

Tests that already need to be carried out in the laboratory include ribonucleic acid (RNA) tests in blood plasma starting on day 7, p24 antigen (Ag) after 10 days, and antibody-based tests (Ab) about 28 days after infection [50]. Using tests that can detect both Ab and p24 reduced the time required. Most RTs are as effective as fourth generation ELISAs [50,51]. A Polish study shows that the only available option for HIV diagnosis in diagnostic laboratories is generally available for the detection of HIV-1 and HIV-2 antibodies, based on the ELISA method, which is currently being carried out in the laboratory [52].

### 3.3. Does TGW Know and Apply HIV Tests?

Despite the large problem of HIV infection, the frequency of HIV testing in the trans population is unsatisfactory [53]. Many TGW postpone medical consultation and come in only with very serious medical conditions [50]. The low rate of HIV testing in TGW [50,54] indicates the need to find new strategies for testing this population. Outreach programs where health services reach out to communities for HIV screening have proven to be an effective service in reaching at-risk populations, including persons providing sexual services [50,55]. TGW who were reached by community-based HIV services and received HIV education were more likely to undergo a recent HIV test [8]. One study found that TGWs who stated they were unlikely to be HIV positive and who always used condoms with non-commercial male partners were less likely to take the test. This study did not capture how or why these characteristics are associated with not testing for HIV. Other studies have found that fear of HIV test results, the associated stigma [8,56], and the tendency among transgender communities to believe that they are not at risk for HIV infection were responsible for non-testing [8].

One of the few studies of European residents found that transgender people face significant barriers to HIV testing services, which may be due to the limited dissemination of HIVST information [29]. In the United Kingdom, nearly 50% of the 500 trans participants admitted that they had never had such a test in their lives [57]. Similar results were found in a study conducted in Thailand, where 53.4% of TGW had never had an HIV test [8,58]. Nonprofit facilities were the most common location to obtain HIV testing and counseling services. This preference confirms previous research in Cambodia, which indicates that community-based services are preferred by TGW and other key groups such as sex workers and MSM [8,59,60,61]. Despite the wide availablity of free HIV testing services, one in five TGWs in the nationwide survey had never taken an HIV test in their lives, and about half had not been tested in the past six months [8]. Similar results were found in a nationwide survey conducted in the United States where 22.8% of respondents had never had an HIV test, with no significant differences between trans male and trans women [8,53].

### 3.4. Motivating and Demotivating Factors for Self-Testing

From the study conducted using web-based information, it was shown that reaching young black TGW with adapted information regarding optimal HIV testing options had a significant impact on HIV testing rates. Once such information was provided, testing rates increased among the study population. These results confirm that interventions are required that actually require minimal effort to inform potential infected persons about testing to disseminate HIV testing [27].

Shresta et al. (2020) studied factors influencing the willingness to undergo HIV self-testing among Malaysian TGWs. Almost half of the participants (47.6%) expressed a willingness to selftest. This willingness, in a multivariate analysis, was positively associated with having experienced sexual assault in childhood, having ever used a cell phone or app to find sex clients, and having had intercourse without condoms in the past six months. Interestingly, it was noted that a willingness to selftest was negatively associated with the presence of depressive symptoms or having more daily clients [62].

The motivation for taking an HIV test is having anal sex without a condom or condom failure. Another motivation is not remembering the details of the last sexual episode, lack of confidence in the partner’s health status, and also having sex without condoms. More frequent willingness to test was observed when there was a large number of sexual partners or frequent sex without condoms. Testing both partners before intercourse without a condom is also a noticeable trend [63]. The above-mentioned motivations for HIVST may apply to both transgender and cisgender people, women and men to the same extent.

The paper by Iribarren et al. describes TGWs’ experiences with HIV self-testing. Participants gave reasons for refraining from HIV testing in the past. The most commonly reported were availability of testing (men lack of knowledge of where and how to get it) and cost. One-fifth of the respondents had never thought about testing for HIV before. The same number of participants stated that they prefer to take the test at a clinic. Other reasons included fear of a positive result, willingness to test with blood, uncertainty of their partner’s reaction to the test result, or a positive test result in their partner [64].

### 3.5. Advantages of Self-Testing

HIVST has been shown to be safe and effective in increasing HIV testing rates without affecting condom use, social protection, or treatment inclusion [65,66]. Home testing provides easy access for high-risk individuals who rarely get tested [13]. Global studies have also shown that no serious side effects associated with HIVST have been reported [41,67,68]. One obstacle to HIV testing is stigma, but the convenience and confidentiality provided by HIST may make it an attractive option for TGW. In addition, individuals who test negative may be associated to other HIV prevention methods, such as pre-exposure prophylaxis [69].

One study found that among those who had sought testing in the past, 47% preferred home testing over testing in a clinical setting. Acceptability of home testing was 90% among those who rarely tested, underscoring the importance of home testing, especially in at-risk populations who may not have access to clinic testing for a variety of reasons [13,70]. There is evidence that HIV testing at higher frequency is related to a lower risk of HIV infection and is an effective regimen for reducing HIV infection rates [71,72,73]. Other studies have found that getting tested before sexual intercourse reinforces sexual decision making, by forgoing or limiting sexual acts or insisting on condom use with HIV-positive people [74,75]. Some studies have shown that strengthening the position is a benefit of HIVST use [74,76,77,78,79]. A large, randomized trial did not show that HIVST use results in better adherence to pre-exposure prophylaxis, yet HIVST was well tolerated and used as a part of prophylaxis for self-testing and testing with sexual partners [12].

A TGW study in New York City and San Juan, Puerto Rico found that of 368 individuals, only 6 (<2%) did not test themselves or their partners. This shows that ST acceptability was extremely high in the study population of interest. Furthermore, study participants showed high motivation to use self-testing, given that not only did 78% of those in the intervention group (they had access to free self-testing) use self-testing kits, but also 13% of participants in the control group (they did not receive self-testing from the organizers) found a way to obtain self-testing. Furthermore, 76% of participants in the intervention group were able to get a potential sexual partner to selftest. This indicates that the dissemination of selftests through networks has important potential to complement and potentially increase the reach of formal outreach programs. Findings suggest that participants’ willingness to use HIVST is highly dependent on predictors of their partners’ reactions to the test proposal [30,80]. Another advantage of sexual partner self-testing was the detection of HIV in individuals who, despite awareness of HIV infection, engaged in unsafe intercourse without first informing their partner about HIV. Sexual partner self-testing also carried with it support in case of a positive result and an attempt to maintain further contact [75]. MSM and TGW participated in a qualitative study conducted in the Philippines and most of them reported that HIST was convenient and time saving. In addition, some informants described HIST as “discreet” and “very confidential” [81]. TGW in particular may associate clinical testing with stigma and lack of privacy [74,76,82], and having HIVST may increase their sense of control over the testing process [74,83].

In the Iribarren study, the main benefits of HIVST reported by participants were an increased sense of security about their partners, ease of taking the test, and assurance of privacy [64]. Those surveyed in Burma, an Asian country, also consider confidentiality and privacy as well as time saving as advantages of HIVST. Moreover, it is a painless method, which was particularly important for people who avoid contact with needles and finger pricks [23]. Most respondents emphasized that being able to test for HIV at home is an advantage and allows them to avoid potentially stigmatizing experiences in health care facilities or the embarrassment associated with test results [31].

A majority (95.7%) of respondents, when asked for their opinions on the HIVSTs they had taken, stated that they were easy to use and much preferred self-testing over testing at a clinic. Almost all those surveyed would recommend HIVST to friends and other interested parties [12]. A 2020 study in which HIV tests were provided to people who identify as TGW points to the role of self-testing in getting HIV-infected people on the right treatment faster and the task of tailoring tests to the specific needs of this particular group so as to achieve higher suppression of HIV transmission in this population [84].

### 3.6. Disadvantages of Self-Testing

Transgender people also had concerns about HIVST, most commonly related to concerns about poor mental health outcomes. Numerous concerns were described about people who find out they are HIV positive and do not seek professional medical help but decide to commit suicide. Concerns were also raised about obtaining uncertain results, which may be due to lack of instruction from qualified persons as to how this test should have been performed and subsequent mis-testing [23]. A frequently observed problem with HIVST in research is the lack of support for those who have received initial positive results. Several study participants feared after receiving a positive result they would be stressed and take drastic actions, including the possibility of self-harm. In addition, some participants felt that it was a disadvantage that they would not automatically receive counselling on HIVST immediately and felt that this lack of counselling could undermine HIV surveillance and could lead to conscious or intentional unprotected sex following information about HIV infection [31]. In Poland, there is the National AIDS Center, which has been operating on behalf of the Minister of Health since 1993. It enables online, anonymous counseling on HIV issues, including providing information on the possibility of HIV testing in a laboratory. HIV tests are available anonymously and free of charge at consultation and diagnostic centers as well as at infectious disease clinics. Taking the test should be combined with talking to an HIV counselor. The study shows, however, that post-test counseling is omitted and performed rarely [52]. A large HIV counseling study found that interest in HIV was low in the Dominican Republic and Georgia (9.1% and 13.8%, respectively) and higher in India (36.8%) [85].

Saliva-based self-testing is less sensitive than other tests in the early stages of HIV infection. Thus, the complete replacement of blood-based HIV testing and saliva-based self-testing may affect the increase in HIV infection [84]. Less than half of the survey participants reported confidence in this diagnostic method, with most considering blood tests to be more reliable. On the other hand, they are more likely to conduct a gum swab together with their partner than a so-called gingival test. “Finger-stick test” in the lab [64].

In TGW’s experience, they might have been exposed to aggression and negative feedback from their sexual partner after being offered an HIV test before intercourse. Therefore, in some cases, it is better to do the test before proposing it to their partners [40]. It has been hypothesized that violent reactions from partners can cause potential reluctance to test. Studies have found several cases of partner violence associated with HIVST. Two studies conducted in Malawi and Kenya in which female participants used HIVST found few incidents of intimate partner violence associated with HIVST [47,86,87]. Another study of female persons providing sexual services in Zambia found only three cases of intimate partner violence among 965 participants [86,88]. TGWs have been shown to be able to successfully use HIV self-testing with sex partners with low rates of violence [86]. Transgender people who were participants in the study conducted by Giguere R et al. recognized that suggesting HIVST to their clients could lead to violent situations. The study participants listed several strategies to minimize the likelihood of violence, such as testing only clients’ friends, using the test in a public place where help would be readily available, and avoiding offering testing to clients who have consumed alcohol or drugs [48,74].

Another limitation in the use of HIVST is that some transgender people avoid introducing unpleasant topics for conversation (e.g., safer sexual behavior) because they fear that they will upset their partners and that it will endanger their relationships [89,90]. Participants in the same study revealed that they did not raise the topic of HIVST with sexual partners when they felt they might be particularly upset or offended [89]. In a 2020 study by Carballo-Diéguez A. et al., among the reasons mentioned by transgender people for whom they did not ask their partners to take the test, there is afear of ending intercourse (25%), the fact that partners will not engage in anal intercourse (24%), and the feeling that the partner may react negatively (19%) [80].

It has been shown that stigmatization of transgender people by doctors in Malaysia can lead to the creation of a hostile environment for TW, including limited access to and use of therapeutic and preventive services [62,91]. Obstacles to self-testing for HIV also include factors such as stigma and discrimination against transgender people and their concerns about confidentiality, as well as limited access to HIV care and a lack of trust in medical facilities [62,92,93].

Another consideration is the high cost of testing. The current market cost of self-testing, which is about $40 in the U.S., would be high for many people, especially considering that two test kits would be required to test each other. There is a need to explore how to lower the cost or provide free kits to people at high risk of HIV infection [80]. Most participants in the study conducted by Giguere R et al. reported that, although they would like to continue using HIVST to screen clients, cost was a significant inhibiting factor. It is too high a cost for transgender women, who may have a large number of sexual clients for whom they would need testing, to use HIVST regularly. Studies have shown that cost is a significant barrier to the use of HIVST [74,76,78,82,83,94], especially as those with the highest prevalence of undiagnosed HIV are often among the lowest socioeconomic status [74,94]. Studies that have assessed willingness to pay for HIV kits among populations at risk for HIV infection have found that most are unable to pay more than $10–20 for a test kit [34,74,76,78].

### 3.7. Future Actions Concerning HIV Self-Testing

Studies have noted the need to provide people at high risk of HIV infection with some form of counselling before providing an HIV selftest kit in order to obtain psychosocial support following a possibly positive test result, and to offer treatment options, in the case of such a result, in a non-stigmatized health care setting [23]. Research on at-risk populations suggests that health care providers should present HIV testing as a routine health check, emphasizing that it should not be a one-time practice. Thus, it is important to encourage clients and patients to make regular visits for HIV testing, for example every three or six months [95]. Healthcare professionals can identify patients who show signs of life chaos, such as difficulty keeping appointments or reluctance to make the next appointment, and offer support services by, for example, scheduling or reminding them of appointments or arranging transportation, thus making health care a source of stability [96].

Participants in the study described by Iribarren indicated a need for a smaller and more manageable package for tests. It was also suggested that such kits should be promoted to vulnerable populations and made more accessible. Another suggestion from the respondents was to enrich the self-testing kits with tests for other diseases [64].

Collecting detailed epidemiological data on transgender women and the number of HIV infections in this population can help identify trends in the HIV epidemic and enable the development of appropriate health plans [97].

### 3.8. Combined HIV and Syphilis Testing

Given the high prevalence of HIV and syphilis among transgender people, it may be a good idea to popularize dual rapid testing for both viruses [63].

It has been noted that there is a correlation between past or active syphilis and HIV infection. Among those with confirmed syphilis infection, the prevalence of HIV was 33.5%. The use of antiretroviral drugs for prophylaxis provides protection against HIV transmission, but not against other sexually transmitted diseases such as syphilis. Syphilis infection can be asymptomatic and remain unnoticed for several years until symptomatic. Therefore, screening and treatment for syphilis and other sexually transmitted diseases with concurrent HIV testing, treatment, and prophylaxis are recommended [24].

INSTI Multiplex are rapid tests that detect HIV and syphilis and are based on a blood test. These tests are patented by bioLytical laboratories and provide very rapid results for HIV and syphilis. This test is available in Canada and Europe, but its use has not yet been approved by the FDA, though the company responsible for the test is about to apply for it. INSTI Multiplex is not currently available as a selftest [63].

## 4. Limitations

The work is limited as there are few reports of HIVST prevalence by the TGW in Europe. At the same time, it notes the need for further research in this direction. The variety of participants and methods used in these studies makes it difficult to compare HIVST prevalence clearly between TGW and other regions of the world.

## 5. Conclusions

TGW are a group at increased risk for HIV infection. HIVST can play an important role in reducing its dissemination and is the preferred method of testing among transgender women due to its confidentiality and convenience. However, it is important to note that there are barriers to HIV testing among TGW. Possible experiences of stigma and discrimination from medical personnel, sexual partners, or society may cause reluctance to self-testing. Through such actions these groups of people can be excluded from the area of medical prevention, which is very unfair. For this reason, there is a need to promote and inform about the possibility of HIVST and to make it more easily accessible and reasonably priced.

## Figures and Tables

**Figure 1 ijerph-19-09331-f001:**
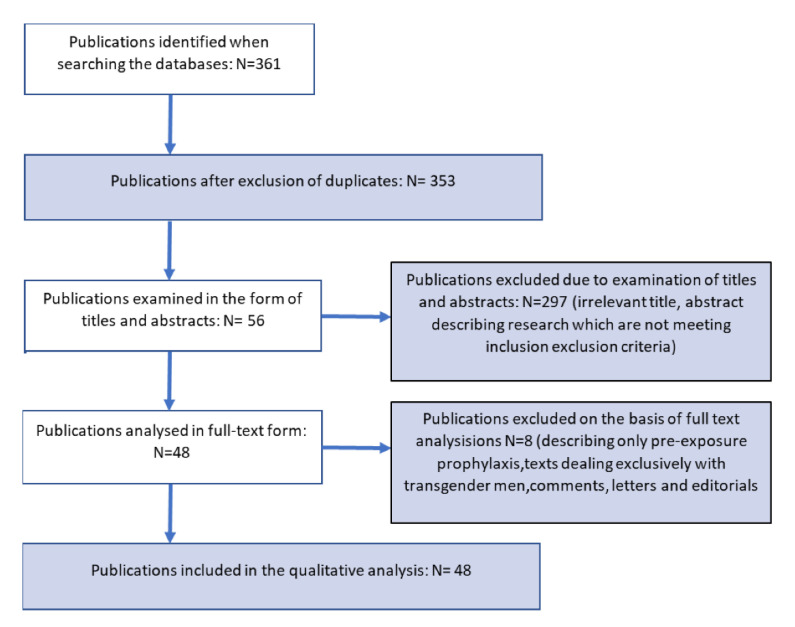
Search strategy.

## Data Availability

Not applicable.

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
