# Peer review of "Self-Testing as a Hope to Reduce HIV in Transgender Women—Literature Review"

_ijerph, 2022, doi:10.3390/ijerph19159331_

Round 1
Reviewer 1 Report
The work concerns an important topic from the point of view of public health, which is the prevention of HIV infection. In the introduction, the authors declared that they used the PRISMA methodology, but did not present its course at any workplace. In particular, the criteria for including and excluding individual articles from the review were not indicated. In this way, the authors very sporadically touch on the issue of HIV self-testing in Europe, and even more so with regard to Polish research, from a region about which - it seems - they would have the most say.
Moving on to a detailed review. At the outset, I would like to point out that the work has no aim and no scientific theses (hypotheses) have been put forward. It is not known why the authors decided to conduct this study at all, why did they focus on transgender women (TGW), is it due to some specific reason or selected literature? The lack of aim is a serious scientific drawback of the work. In the part concerning affiliation, an editing incorrectness was identified - unnecessary space fields. In part, the introduction in line 31, the authors use the non-inclusive term "prostitute", which should not be used in scientific language, is stigmatizing. It would be correct to use the word "sex worker" here, which is generally accepted. In part 2. Materials and methods, the authors refer to the PRISMA methodology but do not state how many articles at which stage of the methodology were excluded and for what reasons. Why is there no reference to European research in the review? The authors completely ignored the availability of self-testing in various regions of the world, e.g. is there validated and certified tests with a similar level of diagnostic reliability in Asia, Europe, USA? Only confirmation of this thesis would allow the regions to be compared with each other. In lines 108-112, the authors continue to think about self-testing by including additional tests, without indicating that they cannot be performed in self-testing mode, but only in a laboratory. The reader may be under the misconception that can also do these results yourself. Part 3.2. concerning the role of the Internet, social media and applications in HIV self-testing is, according to the reviewer, highly theoretical. In practice (especially in Europe and Asia) such applications are only a guide, they do not support the testing, especially self-testing. In addition, they do not offer pre-post-test counseling. The large variety of applications and their functions, according to the reviewer, does not allow for drawing joint conclusions on this subject in this context.
In section 3.3. on motivation and demotivation for self-testing The authors write about numerous factors that make patients to test. But when writing about TGW, they fail to point out that the same criteria are common to all people who perform such tests (especially when it comes to sex without a condom or condom damage). The authors especially (in line 195) indicate that the motivation is anal sex, but this is not the only way of HIV infection, so the type of intercourse does not matter here (anal, vaginal, oral, etc.). In verse 199, the authors suggest that self-testing be promoted prior to intercourse, but doing so in the absence of awareness of the serological window (6 weeks after the risk behaviour) makes no sense. I encourage the Authors to criticize what other authors wrote, because it does not always have to be in line with the current medical knowledge and Evicence Based Public Health. In line 173, the authors noted, in the context of the quality of self-tests, that as many as 100% of the reactive results were confirmed, but that only 52.8% of people were treated with antiretroviral therapy. They don't carry on with that thought. Why is it so, why is there such a large scale of refraining from treatment? How long after the test was this observation made, maybe the patients received treatment later? In section 3.5. Disadvantages of self-testing The authors indicate that one of them is the lack of pre-test counseling. This is a very important problem that, to a large extent, puts self-testing at the end of the list of HIV diagnosis frequency all over the world. Patients are often unable to perform the test correctly, hygienically, do not understand the result, do not know how to behave in the event of a reactive result. Lack of this awareness is a key problem in this matter. The reviewer would expect at this point to present the experiences of some countries (eg in Europe, especially Poland) regarding the use of sexual health clinics, pre-post-test counseling and patient contact with a professional HIV counselor. In section 3.7. The authors discuss the simultaneous testing for HIV and syphilis. A cardinal error was made because lines 326-327 read "... this may be a good idea to popularize double testing for both viruses". Syphilis is not caused by viruses but bacteria - Treponema pallidum spirochetes. Such a mistake is serious in the medical community.
General remarks: the work would be more valuable if the review was statistical in nature, in which the correlation between the results of individual authors could be made, or the obtained results could be presented in tables. Meanwhile, it is a literature description in which the reader gets lost easily. There is a constant lack of aim in the work, and at the end it is not known what the result is. There are no practical conclusions that can be used for further studies or activities. Some conclusions contain too general terms, known to everyone from the common knowledge, that could be formulated also without this research, and thus do not refer to the obtained results.
Reviewer 2 Report
The authors discuss HIV self-testing a way to reduce HIV prevalence among transgender women.
Suggestions for refinement: In the title it would be advisable to replace the word "prevent" (line 2) with "reduce", especially since HIV prevalence cannot realistically be eliminated entirely to zero deaths per 1,000 in a population.
Introduction: It needs to be clear whether the scope of review is worldwide or mainly focused on TGW in Asia. For example, the rationale cited for increased access to HIV testing among TGW is mainly one study in Thailand (lines 59-62).
Methods: the authors state that they have conducted the review in accordance with PRISMA guidelines for systematic review. Generally, the materials methods section could include a figure with a flow diagram of the article selection criteria (screening, exclusion).
Clarification needed - Distinguish between dissemination of testing (Lines 113-140) and dissemination of HIV infection (Line 342).
A stronger connection is needed between the conclusions and the results of the systematic review. For example, in the conclusion section, the authors cite various factors including: experiences of stigma and discrimination form medical personnel, sexual partners, or society as potential barriers to HIV self-testing.
These factors should be discussed in greater depth in earlier sections of the manuscript. Experiences of stigma and discrimination from medical personnel, sexual partners, or society as potential barriers to HIV self-testing- these factors should be discussed in greater depth in earlier sections of the manuscript.
Round 2
Reviewer 1 Report
The authors responded to my comments, although I did not find an answer to question 6. To the remaining words, the answer was concise and essentially did not respond to my objections. I still cannot find an explanation on how the PRISM protocol was used and how many works were qualified for each stage of this protocol, and for what reason it was rejected (the criteria given by the authors are not specified). The aspect in question 8 was intended to draw attention to the deletion of this paragraph as it is not related to the purpose of the paper, rather than to clarify it. In question 11, it was also about supplementing the work with the context of guidance in the works that are described, as well as drawing attention to this aspect. The authors chose a description that is known to the reviewers but does not relate to the work. The authors noted that post-test counselling is very rare (answer 11). The reviewer does not share this opinion unless the Authors have some credible reports on this. I am not submitting any additional comments to this work.
